# Application of Support Vector Machine to Obtain the Dynamic Model of Proton-Exchange Membrane Fuel Cell

**DOI:** 10.3390/membranes12111058

**Published:** 2022-10-28

**Authors:** James Marulanda Durango, Catalina González-Castaño, Carlos Restrepo, Javier Muñoz

**Affiliations:** 1Department of Electrical Engineering, Universidad Tecnológica de Pereira, Pereira 660001, Colombia; 2Centro de Transformación Energética, Facultad de Ingeniería, Universidad Andres Bello, Santiago 7500971, Chile; 3Assistant Investigator Millennium Institute on Green Ammonia as Energy Vector (MIGA), Santiago 7820436, Chile; 4Department of Electrical Engineering, Universidad de Talca, Curicó 3340000, Chile; 5Principal Investigator Millennium Institute on Green Ammonia as Energy Vector (MIGA), Santiago 7820436, Chile

**Keywords:** support vector machine, regression model, proton-exchange membrane fuel cell, diffusive model, evolution strategy, voltage–current dynamic response

## Abstract

An accurate model of a proton-exchange membrane fuel cell (PEMFC) is important for understanding this fuel cell’s dynamic process and behavior. Among different large-scale energy storage systems, fuel cell technology does not have geographical requirements. To provide an effective operation estimation of PEMFC, this paper proposes a support vector machine (SVM) based model. The advantages of the SVM, such as the ability to model nonlinear systems and provide accurate estimations when nonlinearities and noise appear in the system, are the main motivations to use the SVM method. This model can capture the static and dynamic voltage–current characteristics of the PEMFC system in the three operating regions. The validity of the proposed SVM model has been verified by comparing the estimated voltage with the real measurements from the Ballard Nexa® 1.2 kW fuel cell (FC) power module. The obtained results have shown high accuracy between the proposed model and the experimental operation of the PEMFC. A statistical study is developed to evaluate the effectiveness and superiority of the proposed SVM model compared with the diffusive global (DG) model and the evolution strategy (ES)-based model.

## 1. Introduction

The faster development and integration of renewable energies and electromobility technologies in the distribution grid have encouraged the necessity to find alternatives for power sources. One of the ways of decarbonization is the use of hydrogen. The hydrogen fuel cell consists of an electrochemical device that directly and continuously converts hydrogen fuel to electrical energy directly and constantly. This device is essential to tackle climate change and air pollution and switch to green energy. Within the different types of FCs, the PEMFC is considered the most exciting power source for many applications, such as transportation, mobile devices, heat, and power supplies for buildings and industry [1]. Its many advantages include low operating temperature, high power density, fast startup, quick response, and zero emissions. Due to its benefits and applications, industry and academia have conducted research and exploration to optimize the performance of the PEMFC. The PEMFC is a nonlinear, multi-variable and very complex system [2]. Due to its complicated process, a control system with good performance is required to regulate the voltage. Therefore, many researchers have focused their interest on the modeling study of PEMFCs.

The models developed not only describe the physicochemical phenomena in the fuel cells but also generate data-driven models based on the functional relationship between the current and the voltage of fuel cells [3]. The process of a fuel cell is usually characterized by the current–voltage curve, known as the polarization curve. There are three main types of polarization: activation polarization, ohmic (or resistance) polarization, and concentration polarization. The deviation of the cell potential from the ideal behavior is a direct result of the sum of these factors over the entire load range. Fuel cells have the best performance in the ohmic polarization region [4].

The FC model comparison based on artificial intelligence and electrical circuit models is shown in Table 1. The models presented in [1,3,5,6,7,8,9,10,11,12,13] are not viable for analyzing the complete polarization curve of an FC, owing to only considering a linear steady-state response based only on the PEMFC ohmic polarization curve, ignoring the voltage–current dynamic response. On the other hand, most models contemplate an analytical approach based on equations that describe the physical system of the FC [3,9,10,14,15,16,17,18]. Therefore, these approaches need several variables to evaluate the model, such as the operating absolute temperature of the fuel cell ($T_{fc}$), the operating current of the fuel cell ($i_{fc}$), the partial pressures of hydrogen and oxygen at the input channels of the fuel cell stack ($P_{H2}$ and $P_{O2}$), and the resistance of the membrane surface ($R_{m}$). However, due to implementation expenses, using multiple variables for its evaluation increases the development costs and the requirements of the high-processing device. Based on the above, data-driven models (or black box models) emerge as a promising strategy for modeling multivariate, nonlinear, and complex systems such as the PEMFC. Among the data-based models is the SVM. The main advantages of the SVM are its stability, robustness, generality, flexibility in determining the model structures, training capacity with limited data, and good predictive performance [13,19,20]. Therefore, this paper proposes a PEMFC model based on SVM, considering the current as the only input variable to the model, and the voltage as the output. The SVM is used to estimate the non-linear V-I dynamic characteristics of a PEMFC system in all regions.

Several papers related to the PEMFC model using an SVM have been found in the state-of-the-art review. In [13], the performance prediction of the PEMFC system of a commercially available electrical bicycle was investigated using an SVM and an artificial neural network. The results showed that the prediction performance of a PEMFC using an SVM is more effective than an artificial neural network. In [3], the authors proposed a hybrid model by combining a support vector machine (SVM) model with an empirical equation. The SVM model was used to predict PEM fuel cell voltages showing high accuracy even when the operating variables vary. In [21], a comparative study of random forest and support vector regression algorithms is presented to estimate the voltage of a solid oxide fuel cell. The results showed that the support vector regression is better than the random forest algorithm. However, the main drawback of the support vector regression algorithm is the small number of samples used in the validation phase. In [14], a Hammerstein model of a PEMFC stack is built using an SVM. The use of an SVM avoids complicated differential equations to describe the dynamic characteristics of a PEMFC stack, and it is appropriate to model the PEMFC compared to traditional methodologies. Finally, in [2] a review of the applications and contributions of machine learning algorithms in optimizing the PEMFC performance is presented. According to the review, machine learning algorithms such as SVMs have shown higher performance in constructing the data-driven model of the PEMFC. In brief, according to the review of the state of the art, most of the models based on machine learning do not consider the dynamic characteristic of the polarization curve, likewise, its validation is not performed with real FC data, and they have a high cost in their implementation. The proposed model overcomes these disadvantages with high estimation accuracy. The main contributions of this paper are the following:The proposed model has been able to model the static and dynamic characteristics of the polarization curve in all operating regions. Two different FC current profiles have been used, with the aim to evaluate the generalization of the proposed model to predict the fuel cell voltage under different operating conditions.We provide a method to model the PEMFC based on SVM, considering a significant reduction in the number of samples used in the training phase, compared with the number of samples used in the training phase of the models proposed in [18,22]. In the same way, the number of samples used in the validation phase is much higher than the number of samples used in the validation phase of the model proposed in [21].Real measurements were used in the training and validation phase of the SVM model. The data correspond to a commercial Nexa fuel cell power module, with a rated power up to 1.2 kW.The proposed model is accurate. The results showed a high similarity between the voltage predictions obtained by the SVM model and the actual data, obtaining root mean squared errors (RMSEs) of less than 1%. Likewise, the root mean squared error obtained with the proposed model is 62% lower than the evolution strategy [18] and the diffusive model [22]. Therefore, the obtained results prove the effectiveness of the proposed FC model compared with other models.

This paper is organized as follows: Section 2 presents a short description of the concepts related with the SVM. The description of the proposed model is presented in Section 3. The validation of the proposed model is present in Section 4, and finally, conclusions are given in Section 5.

## 2. A Multi-Output Support Vector Machine

This method was proposed in [23]. The authors use a multi-output support vector regressor (M-SVR) to refer to their method. Let $x\in R^{d}$ be the input vector and $y\in R^{Q}$, the output vector. The relationship between x and y is assumed to follow
(1)$$\begin{array}{cc} y & =W^{⊤}\phi \left(x\right)+b, \end{array}$$
where $W=\left[w^{1},\ldots ,w^{Q}\right]$, $b={\left[b^{1},\ldots ,b^{Q}\right]}^{⊤}$, with a vector $w^{j}$ and a constant $b^{j}$ for every output (j=1,…,Q). The function ϕ(·) refers to a non-linear transformation to a higher-dimensional space H, where H≫d.

First, given a dataset ${\left\{x_{i},y_{i}\right\}}_{i=1}^{n}$ that is used to build the mapping from x to y, the purpose is to find the set of parameters {W,b} that minimize the functional
(2)$$\begin{array}{cc} J(W,b) & =\frac{1}{2}\sum_{j=1}^{Q}{∥w^{j}∥}^{2}+C\sum_{i=1}^{n}L\left(u_{i}\right), \end{array}$$
where *C* is a regularization constant, and $L(u_{i})$ is known as the Vapnik ϵ− insensitive loss-function, with $u_{i}=\sqrt{e_{i}^{⊤}e_{i}}$, and $e_{i}=y_{i}-W^{⊤}\phi \left(x_{i}\right)-b$. As a loss function L(·), the authors of [23] use a differentiable quadratic loss with respect to a user-defined constant, ϵ, that takes into account all outputs to obtain each individual regressor $w^{j}$ and $b^{j}$
(3)$$\begin{array}{cc} L(u_{i}) & =\left\{\begin{array}{cc} 0, & u_{i}<\epsilon \\ {\left(u_{i}-\epsilon \right)}^{2}, & u_{i}\geq \epsilon . \end{array}\right. \end{array}$$

The solution to the above-mentioned problem can also be expressed in terms of the vector of coefficients $\beta^{j}$ for each output, which relates to the original vectors $w^{j}$ through $w^{j}=\Phi^{⊤}\beta^{j}$, where $\Phi ={\left[\phi \left(x_{1}\right),\ldots ,\phi \left(x_{n}\right)\right]}^{⊤}$. The problem to be solved can then be written as (see Ref. [23])
(4)$$\begin{array}{c} \left[\begin{array}{cc} K+D_{a}^{-1} & 1\\ a^{⊤}K & 1^{⊤}a \end{array}\right]\left[\begin{array}{c} \beta^{j}\\ b^{j} \end{array}\right]=\left[\begin{array}{c} y^{j}\\ a^{⊤}y^{j}, \end{array}\right] \end{array}$$
where K is a kernel matrix with entries $k(x_{i},x_{i}^{\prime })$ computed from a so-called kernel function $k(x,x^{\prime })$. The kernel function generalizes the inner product between ϕ(x) and $\phi (x^{\prime })$. (In the sense that the kernel is not necessarily a Mercer kernel, this is one that can be written as an inner product. See [24].) Additionally, in expression (Equation 4), $a={\left[a_{1},\cdots ,a_{n}\right]}^{⊤}$, and $D_{a}$ is a diagonal matrix with entries ${\left\{a_{i}\right\}}_{i=1}^{n}$. The terms ${\left\{a_{i}\right\}}_{i=1}^{n}$ are computed using [23]
(5)$$\begin{array}{c} a_{i}=\left\{\begin{array}{cc} 0, & u_{i}<\epsilon \\ \frac{2C(u_{i}-\epsilon )}{u_{i}}, & u_{i}\geq \epsilon \end{array}\right. \end{array}$$
where $u_{i}$ was defined before.

An iteratively reweighted least squares (IRLS) procedure is proposed in [23] to solve (Equation 4), in order to calculate the parameters $\beta =[\beta^{1},\ldots ,\beta^{Q}]$ and b.

Once we calculate the matrix β and the vector b, the prediction $\hat{y}$ for a new input vector x can be computed as
(6)$$\begin{array}{c} \hat{y}=\beta^{⊤}k_{x}+b, \end{array}$$
where k is a vector with entries given by ${\left\{k\left(x_{i},x\right)\right\}}_{i=1}^{n}$. The kernel function $k(x,x^{\prime })$ that is used in this paper is the radial basis function (RBF) kernel, given as
(7)$$\begin{array}{cc} k(x,x^{\prime }) & =\exp\left\{-\frac{1}{2\sigma^{2}}{\left(x-x^{\prime }\right)}^{⊤}\left(x-x^{\prime }\right)\right\}, \end{array}$$
where the parameter $\sigma^{2}$ is usually known as the bandwidth [24].

## 3. Proposed Model of the PEMFC Based on SVM

The proposed method is divided into two parts called the training phase and the validation phase. In the training phase, the matrix β and the vector b of the SVM are calculated, taking as input the real data of current and voltage from the Nexa® FC power module. These data are called training samples. A conceptual illustration of the SVM training phase is shown in Figure 1.

With reference to Figure 1, the minimization block performs the calculation of β and b by the minimization of the error function $e_{v}$ between the measured *v* and the estimated $\hat{v}$ voltage using the set of the training samples. This minimization is formulated as an iterative procedure given by
(8)$$\begin{array}{c} \left[\begin{array}{c} \beta^{p+1}\\ {\left(b^{p+1}\right)}^{⊤} \end{array}\right]=\left[\begin{array}{c} \beta^{p}\\ {\left(b^{p}\right)}^{⊤} \end{array}\right]+\eta^{p}\left[\begin{array}{c} \beta -\beta^{p}\\ {\left(b-b^{p}\right)}^{⊤} \end{array}\right], \end{array}$$
where $\beta^{p}$ and $b^{p}$ are the values at the iteration *p*. The value of $\eta^{p}$ is computed using a backtracking algorithm, in which, if $J^{p+1}>J^{p}$, then $\eta^{p}$ is multiplied by a positive constant less that one, and the regressors $\beta^{p+1}$ and $b^{p+1}$ are computed again, until a decrease is achieved in $J^{p+1}$. The algorithm stops when $J^{p+1}-J^{p}<$ tolerance. In this paper, the tolerance is set to $10^{-9}$ [23]. It is worth noting that the training phase is performed offline and it can take several seconds.

In the validation phase, the estimation of the FC voltage is performed from the validation measurements. In an online estimation, a sample of voltage is estimated for each current sample. Connecting with the description given for the SVM in Section 2, d=1 and Q=1 (i.e., x and y are scalars). Taking into account the above, (Equation 6) can be re-written as
(9)$$\begin{array}{c} \hat{v}=\beta^{⊤}k_{i}+b, \end{array}$$
where β is a column vector with $N_{t}$ rows, $N_{t}$ being the number of samples considered in the training phase, and *b* being a constant value. According to (Equation 9), the estimated voltage samples $\hat{v}$ are calculated given the vector β, the scalar *b* (both calculated in the training phase), and the measured current sample *i*. The kernel vector $k_{i}$ is computed using (Equation 7) using the current vector $i_{t}={\left[i_{1},\cdots i_{k},\cdots i_{N}\right]}^{⊤}$ whose entries are the training samples of current, and a sample of real current *i*. In brief, for each sample of the current *i*, one sample of voltage is estimated, in this way, for *N* samples of the currents is formed a voltage vector whose entries are the estimated samples of voltage $\hat{v}={\left[{\hat{v}}_{1},\cdots ,{\hat{v}}_{k},\cdots ,{\hat{v}}_{N}\right]}^{⊤}$.

## 4. Experimental Results

### 4.1. Dataset Measured from the PEMFC

In this work, real data taken from the Nexa® fuel cell have been used, which produces unregulated DC power, up to 1.2 kW, from a hydrogen and air supply. LabVIEW software, which supplies a graphical user interface to the operational status and performance of the Nexa module [25], is provided by the Nexa power module. The experimental Nexa PEMFC data acquisition configuration can be observed in Figure 2, employed for training and verification of the SVM model.

To accomplish rapid acquisition, data saving on its onboard hard drive, and long capture time, the LeCroy WaveSurfer 64Xs-A oscilloscope has been employed. The oscilloscope instantly acquires and keeps the data corresponding to the fuel cell current and voltage signals. Furthermore, a virtual instrument is designed through LabVIEW to create the current profiles via the DC electronic load control, utilizing its GPIB communication port as a constant current load. Thus, the maximum sampling limitation of the Nexa software is bypassed, performing sampling periods up to 20 μs.

Two FC load current profiles have been considered with the aim of testing the dynamic of the SVM model. Each profile contains the training and validation sets with different measured data. The first one was used in [22] to train and validate the diffusive global (DG) model. The second one was used in [18] to evaluate a PEMFC model based on an equivalent circuit, whose parameters were identified using an evolution strategy (ES) algorithm, and also used in [22] to evaluate the performance of the DG model.

### 4.2. Training Phase of the SVC Model with the First FC Load Current Profile

Figure 3 shows the instantaneous measurements of the current and voltage of the FC used to select the training samples of the SVM model. The data were sampled with a fixed step time of 200 ms for one hour. A total of 179,988 current and voltage samples were obtained. The number of samples used for training has been substantially reduced to 9000 (5% of the initial data), taking samples every 2000 ms and considering only 50% of the rest of the data, to reduce the size of the vector β of the SVM model given by (Equation 9), which is appropriate to reduce the computational cost in the validation phase. A total of 50% of the samples were randomly selected.

The hyperparameters of the SVM were set to the following values: C=200, ϵ=0.001, and σ=1.5. These values were adjusted according to numerical experiences in the training phase of the SVM and showed a good predictive performance of the model. The root mean square error (RMSE) index defined by (Equation 10) has been used to quantify the performance of the SVM model.
(10)$$\mathrm{RMSE}=\sqrt{\frac{\sum_{k=1}^{N}e_{k}^{2}}{N}},$$
where $e_{k}$ is the error between the measured and the estimated FC output voltages and *N* is the number of samples of the discrete signal $\hat{v}={\left[{\hat{v}}_{1},\cdots ,{\hat{v}}_{k},\cdots ,{\hat{v}}_{N}\right]}^{⊤}$. The RMSE obtained for the training samples is 0.3 V. The authors consider that this value confirms a satisfactory predictive performance in SVM training. Figure 4 shows the relative error between the measured and the estimated voltages, arranged in ascending order. As can be seen, the SVM model presents a good predictive performance due to 99% of the data presenting relative errors of less than 3%.

### 4.3. Validation of the SVM Model with the First FC Load Current Profile

The validation of the proposed model is carried out by the measurements shown in Figure 5, which is different from the one used during the training phase. This current profile is much more demanding than the one used for the validation stage because it has current step changes that are higher in magnitude. The validation data have a fixed time step of 200 ms and have been measured during a 20 min time window. Estimated voltage samples are calculated using Equation (Equation 9), where β is 9000×1 vector and b=32.2729 V. The kernel vector $k_{i}$ is computed using (Equation 7), with x being the vector of training samples of the FC load current and $x^{\prime }$ the validation sample from the FC load current. In a real implementation, each estimated voltage sample is computed from each real current sample. Therefore, the entries of the vector $k_{i}$ must be updated for each sample. The dimension of $k_{i}$ is 9000×1.

The results obtained with the SVM model are compared with the results of the diffusive model presented in [22], using the following indices: R-square, relative error (RE), mean absolute error (MAE), and standard deviation (SD). The mathematical definitions of these indices are presented in Appendix A. The statistical analysis in Figure 6 shows that the proposed SVM model has a low error and a high R-square value compared with the diffusive global model.

Figure 7 shows the dynamic V-I characteristic of the PEMCF using real FC voltage and estimated FC voltage from the SVM model. The maximum difference between the real and estimated voltage is 3.86 V. It occurs when the FC current is 38.61 A. Note that the SVM model can capture the fuel cell’s dynamic characteristic, particularly at the extremes of the curve.

### 4.4. Training of SVM Model with the Second FC Load Current Profile

Figure 8 shows the measurements used to select the set of training measurements for the SVC model, which correspond to the second FC load current profile. The data were sampled within a period of 200 ms in [18], which is 10 times higher than the sampling in the profile in Figure 3. However, to reduce the size of the vector β in (Equation 9), the data used to train the SVM were re-sampled within a period of 2000 ms, and then 50% of the data were taken randomly to train the SVM. The total number of samples used to train the SVM model was 2957.

According to numerical experience, the values of the hyperparameters of the SVM were adjusted to C=5000, ϵ=0.001, and σ=4. The predictive performance of the SVM model was evaluated in terms of the RMSE index. For the training measurements, the RMSE obtained is 0.38 V. The authors consider that the predictive performance of the SVM model is satisfactory according to the obtained RMSE value. The relative error was calculated for all the samples used in the training. According to the obtained results, 97% of the relative errors calculated (i.e., 2868 values) are less than 3%.

### 4.5. Validation of SVM Model with the Second FC Load Current Profile

Figure 9 shows the results of the validation test. Notice the wide range of values considered for the FC load current in the upper part of Figure 9. The estimated samples of the FC voltage are calculated with (Equation 9), taking as input the vector β (2927×1 vector), the constant b=23.17 V, and the kernel vector $k_{x}\left(x,x^{\prime }\right)$ (2927×1 vector) calculated with (Equation 7) using the training samples of the input vector x and the validate sample $x^{\prime }$ of the FC load current. The results were compared with the DG model proposed in [22] and the equivalent circuit of the PEMFC proposed in [22]. The results are shown at the bottom of Figure 9. The RMSE value was calculated for all the models. The RMSE of the DG model and the parameters adjusted by ES were 2.3273 V and 2.3116 V, respectively. Meanwhile, the RMSE for the SVM approach is 0.87 V. This value is equivalent to 37% of the RMSE obtained with the other approach.

The statistical results of the proposed SVM model are detailed in Figure 10. These results are compared with those from the diffusive global model and those from the ES approach. The SVM model has the lowest error value, as it accurately represents the static and dynamic current–voltage relation in a PEMFC.

Finally, Figure 11 shows the dynamic V-I characteristic of the PEMFC using the second profile of the FC load current. The horizontal axis corresponds to the validation samples of the current, and the vertical axis corresponds to the experimental and estimated samples of the voltage. Notice that the SVM model is able to capture the dynamic characteristic of the PEMFC. The maximum difference between the experimental and estimated voltage is 3 V and occurs when the current is 16.5 A.

## 5. Conclusions

In this paper, an accurate and efficient support vector machine-based model has been developed for proton-exchange membrane fuel cells. The important contribution of the proposed modeling approach is the considerable reduction of the samples used in the training phase of the SVM model, compared with other recent PEMFC modeling mechanisms, such as the diffusive global model and evolution strategy-based model. This reduction in the number of samples required in the training phase reduces the dimension of the model vectors and, therefore, reduces the computation time in the validation phase. Two FC load current profiles have been considered to evaluate the predictive performance of the proposed PEMFC model based on an SVM. The real measurements were taken from the Nexa® power module with a DC power up to 1.2 kW. The performance of the proposed model was carried out by computing the RMSE between the estimated voltage samples and the measured ones. The RMSE obtained for the first profile was 1 V, and for the second profile was 0.87 V. These results are lower than those obtained with the diffusive model, which are 1.05 V for the first profile and 2.32 V for the second profile, and the evolution strategy model, which is 2.31 V for the second profile. Notice that, for the second profile, the error obtained with the proposed model is 62% lower than with the other models. According to the performance indices, the predictive performance of the proposed SVM model is better than the DG and ES models.

Once the methodology has been validated, it would be possible to use the SVM model to predict the voltage of the PEMFC. Results show that the proposed SVM model is able to capture the dynamic characteristic of the PEMFC, validating the use of the SVM model in further real applications related to its control systems. In future work, it is recommendable to speed up the convergence in the training process by optimizing the amount of data used.

## Figures and Tables

**Figure 1 membranes-12-01058-f001:**
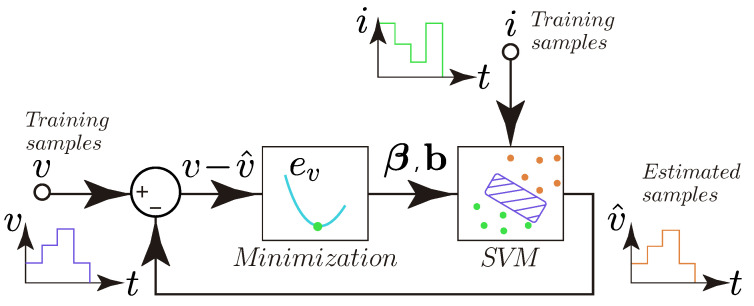
Conceptual illustration of the SVM training.

**Figure 2 membranes-12-01058-f002:**
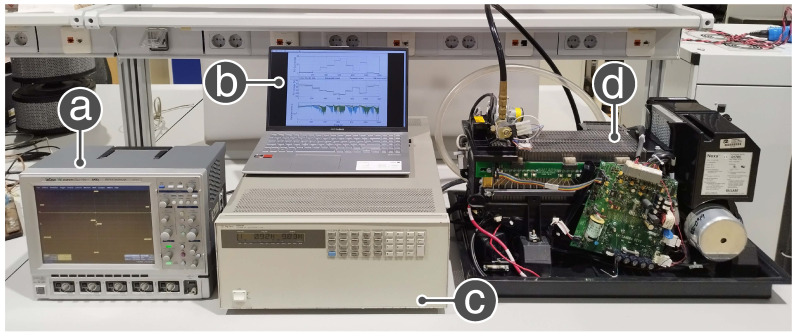
Experimental data acquisition configuration used for the phases of training and validation of the SVM model: (**a**) LeCroy WaveSurfer 64Xs-A oscilloscope, which saves trace data to an internal memory location, (**b**) laptop computer + LabVIEW program to control the electronic load using the National Instruments GPIB-USB-HS+ software program to monitoring the Nexa Fuel Cell, (**c**) Agilent 6050A Electronic load, and (**d**) Nexa power module from Ballard Power Systems.

**Figure 3 membranes-12-01058-f003:**
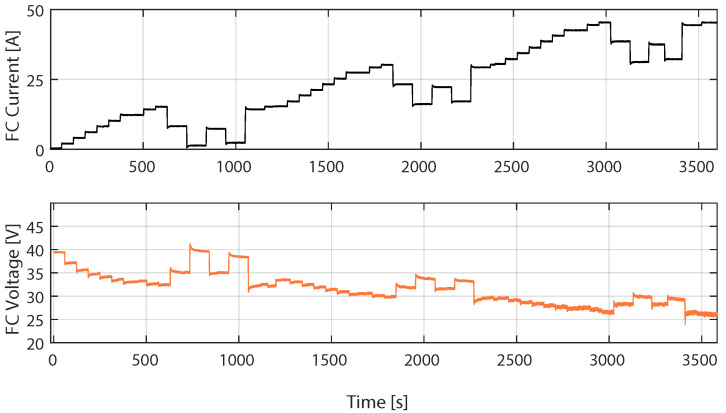
Real data of the first FC load current profile used to select the training samples of the SVM model.

**Figure 4 membranes-12-01058-f004:**
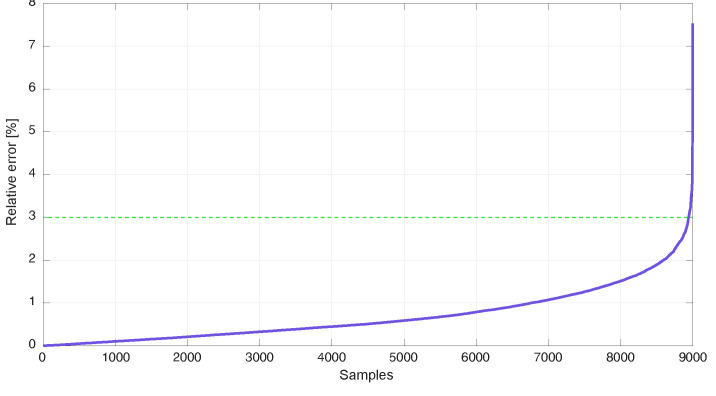
Relative error for each one of the estimated voltage samples used in the training phase of the SVM model.

**Figure 5 membranes-12-01058-f005:**
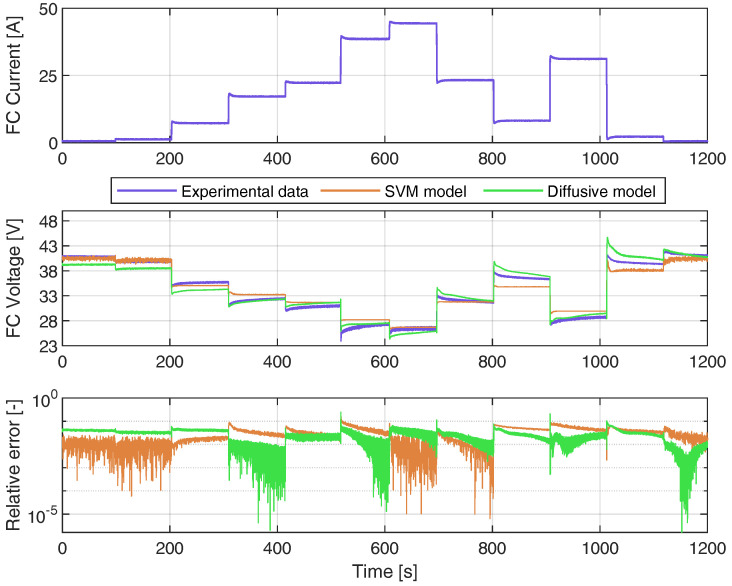
Results obtained from the voltage estimation using the proposed SVM model, (**above**) model input: FC current, (**medium**) real and estimated voltage from the SVM model and diffusive model, and (**below**) relative error between the real and estimated voltage from the SVM and diffusive models.

**Figure 6 membranes-12-01058-f006:**
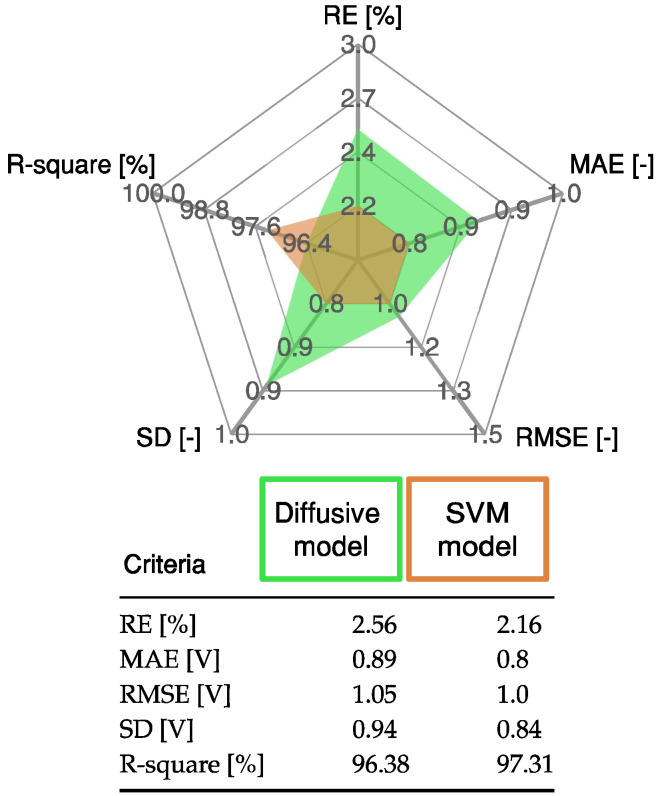
Statistical results of proposed SVM approach and the diffusive global model for the profile shown in Figure 5.

**Figure 7 membranes-12-01058-f007:**
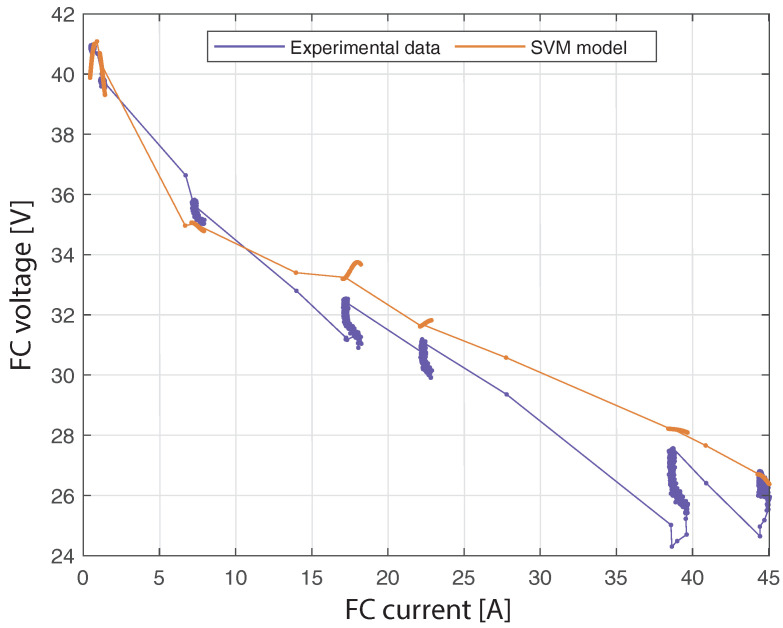
Dynamic V-I characteristic of the PEMFC obtained with real and estimated FC voltage using the SVM model, from the first profile of the FC load current.

**Figure 8 membranes-12-01058-f008:**
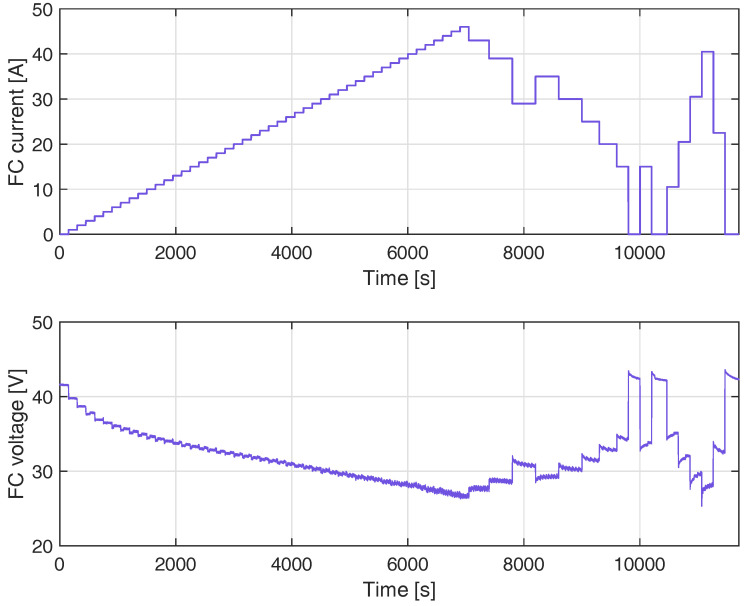
Measurements from the second FC load current profile, used to select the training samples of the SVM model.

**Figure 9 membranes-12-01058-f009:**
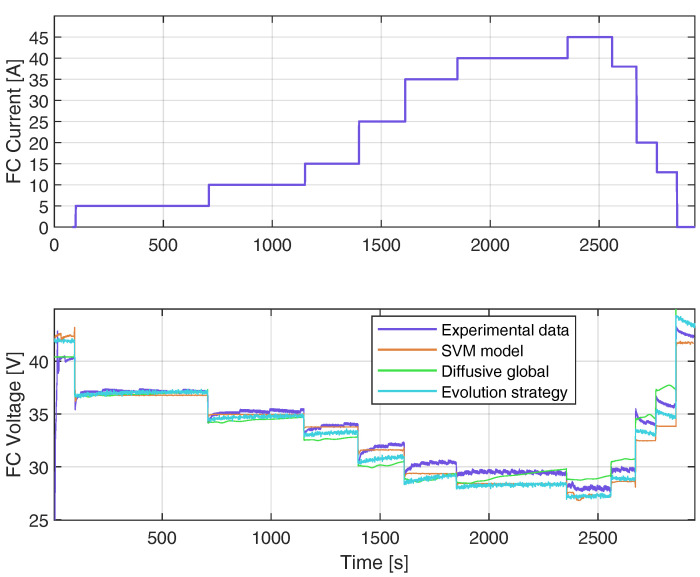
Real data used for validating: current load profile (**above**), and output voltage simulated with parameters estimated by means of ES, the diffusive global model, and the SVM approach (**below**).

**Figure 10 membranes-12-01058-f010:**
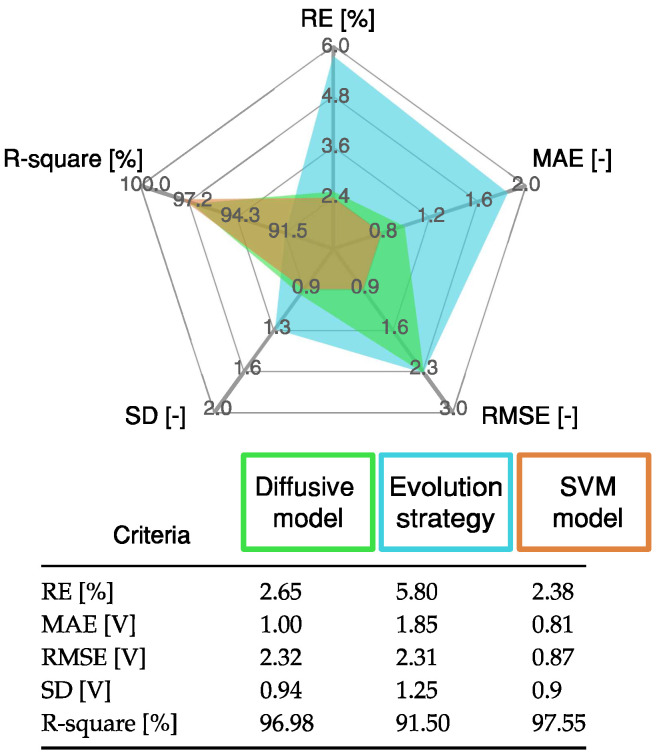
Statistical results of proposed SVM approach, diffusive global model, and ES approach for the profile shown in Figure 9.

**Figure 11 membranes-12-01058-f011:**
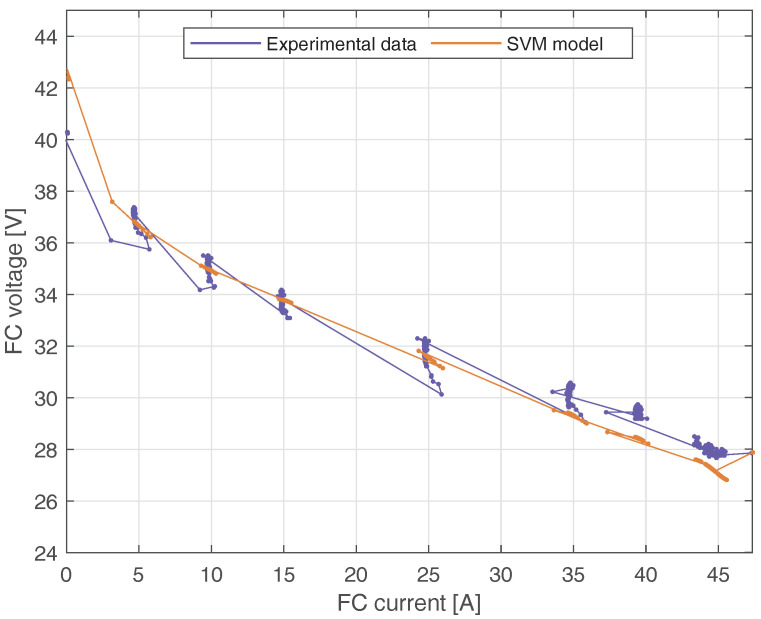
Dynamic V-I characteristic of the PEMFC obtained with real and estimated FC voltage using the SVM model, from the second profile of the FC load current.

**Table 1 membranes-12-01058-t001:** Fuel cell intelligence model comparison.

FC Model Strategy	Ref.	Static Model	V-I Dynamic Model	Variables Used to Evaluate the Model	Training Complexity	Implementation Cost	Tested with a Real FC
SVM PEMFC	[12]		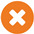	Tfc, ifc	M	H	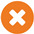
SVR	[13]		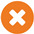	ifc	M	H	
ABC	[1]		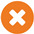	ifc	M	M	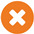
Hybrid SVM	[3]		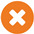	Tfc, ifc, PH2, PO2, Rm	M	H	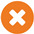
MIMO SVM-ARX	[14]			Tfc, ifc, PH2, PO2, Rm	H	H	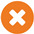
VSDE	[9]		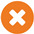	Tfc, ifc, PH2, PO2, Rm	M	H	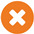
ASO	[10]		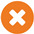	Tfc, ifc, PH2, PO2, Rm	H	H	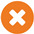
ARX-RLS	[15]			Tfc, ifc, PH2, PO2, Rm	L	H	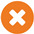
Electrical model	[16]			Tfc, ifc, PH2, PO2, Rm	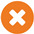	H	
Electrical circuit	[17]			Tfc, ifc, PH2, PO2, Rm	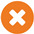	H	
ES	[18]			Tfc, ifc, PH2, PO2, Rm	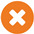	H	
Diffusive model	[22]			ifc	H	M	
This work	[-]			ifc	M	M	

## Data Availability

Not applicable.

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
