# Peer review of "Application of Support Vector Machine to Obtain the Dynamic Model of Proton-Exchange Membrane Fuel Cell"

_membranes, 2022, doi:10.3390/membranes12111058_

Round 1
Reviewer 1 Report
A precise and efficient model of proton exchange membrane fuel cell (PEMFC) based on support vector machine (SVM) is proposed in this paper. This topic is very interesting, and the manuscript is well written. It is recommended to accept. However, it is suggested to add the comparative analysis between your model and other models, and it is necessary to explain the advantages of your work.
Author Response
Reviewer 1
“A precise and efficient model of proton exchange membrane fuel cell (PEMFC) based on support vector machine (SVM) is proposed in this paper. This topic is very interesting, and the manuscript is well written. It is recommended to accept. However, it is suggested to add the comparative analysis between your model and other models, and it is necessary to explain the advantages of your work”.
Answer. Thanks for your comment. In the new version of the manuscript, a comparative analysis between the proposed model and other models proposed in the literature has been carried out. Specifically, a comparative table has been presented in the “Introduction”, highlighting the relevant characteristics of the proposed method, which are not exhibited by the models found in the review of the state of the art. Likewise, the following advantages of the proposed model have been added in the “Introduction”:
- The proposed model has been able to model the static and dynamic characteristic of the polarization curve in all operating regions. Two different FC current profiles has been used, with the aim to evaluated the generalization of the proposed model to predict the fuel cell voltage, under different operation
- Provide a method to model the PEMFC based on SVM, considering a significant reduc- tion in the number of samples used in the training phase, compared with the number of samples using in the training phase of the models proposed in [18] and [22]. In the same way, the number of samples used in the validation phase is much higher than the number of samples used in the validation phase of the model proposed in [21].
- Real measurements were used in the training and validation phase of the SVM The data correspond to a commercial Nexa fuel cell power module, with a rated power up to 1.2 kW.
- The proposed model is The results showed a high similarity between the voltage predictions obtained by the SVM model and the actual data, obtaining root mean squared errors (RMSE) less than 1%. Likewise, the root mean squared error obtained with the proposed model is 62 % lower than the evolution strategy [18] and the diffusive model [22]. Therefore, the obtained results prove the effectiveness of the proposed FC model compared with other models.

Reviewer 2 Report
This paper studies the application of support vector machine in the dynamic modeling of proton exchange membrane fuel cell, which has certain engineering application value. However, in order to meet the high standard publishing requirements of journals, the following suggestions need to be considered.
1) Quantitative data is required, and the highlights are not prominent;
2) The highlights of this article are not given in the first section;
3) The second section is about the existing knowledge. Is it necessary to simplify it? Please consider carefully.
4) Section III also needs to consider this.
5) The paper lacks discussion. For example, compared with the existing literature, the innovation of this paper has not been well reflected.
6) Conclusion There is no quantitative data.
7) Why are there two blank pages? Please check other errors carefully.
Author Response
Reviewer 2
This paper studies the application of support vector machine in the dynamic modeling of proton exchange membrane fuel cell, which has certain engineering application value. However, in order to meet the high standard publishing requirements of journals, the following suggestions need to be considered.
1. “Quantitative data is required, and the highlights are not prominent”.
Answer. We would like to thank the reviewer for his constructive and valuable comments. The highlights has been rewritten in the new version of the manuscript. Likewise, the percentage reduction in the root mean square error with respect to other methods proposed in the literature has been quantified. In the results section, several error indices are presented, which are used to quantify the performance of the proposed method and to compare it with other models. The error indices used are as follows:
- Relative error (RE).
- Mean absolute error (MAE).
- Root mean squared error (RMSE).
- Standard deviation (SD).
- R-squared.
2. “The highlights of this article are not given in the first section”.
Answer. Thanks for your comment. After analyzing the disadvantages of the models proposed in the literature, the following highlights of the article are presented in the intro- duction, which attempt to overcome some of the disadvantages of the models proposed in the literature:
- The proposed model has been able to model the static and dynamic characteristic of the polarization curve in all operating regions. Two different FC current profiles has been used, with the aim to evaluated the generalization of the proposed model to predict the fuel cell voltage, under different operation
- Provide a method to model the PEMFC based on SVM, considering a significant reduc- tion in the number of samples used in the training phase, compared with the number of samples using in the training phase of the models proposed in [18] and [22]. In the same way, the number of samples used in the validation phase is much higher than the number of samples used in the validation phase of the model proposed in [21].
- Real measurements were used in the training and validation phase of the SVM The data correspond to a commercial Nexa fuel cell power module, with a rated power up to 1.2 kW.
- The proposed model is The results showed a high similarity between the voltage predictions obtained by the SVM model and the actual data, obtaining root mean squared errors (RMSE) less than 1%. Likewise, the root mean squared error obtained with the proposed model is 62 % lower than the evolution strategy [18] and the diffusive model [22]. Therefore, the obtained results prove the effectiveness of the proposed FC model compared with other models.
3. “The second section is about the existing knowledge. Is it necessary to simplify it? Please consider carefully”.
Answer. Thank you for the recommendation. The second section “A multi-output support vector machine” presents a short description of the necessary concepts to understand the SVMs of multiple-inputs and multiple-outputs. Also, this section presents the notation used in the article, which is important to facilitate the reader’s understanding of the results obtained. We believe that further simplification of this section may result in the loss of important information for the reader.
4. “Section III also needs to consider this”.
Answer. T Thank you for the recommendation. In section III “Proposed model of PEMFC based on SVM”, the application of the SVM to model the fuel cell is explained. It is necessary to explain the training and validation phases of the SVM for a better understanding of the article. The inputs and outputs of the SVM are defined based on the measurement data of the fuel cell. The dimension of the matrix β and the vector b related with the SVM model are defined. The conceptually illustration of the training phase is given using a block diagram, and a short description of the minimization process is presented. In the description of the validation phase are explained how to use the SVM to predict the samples of voltage of the fuel cell taking as input the current. We believe that the description presented for the proposed method provides the basic concepts necessary for a better understanding of the article, and further simplification may lead to the loss of important information.
5. “The paper lacks For example, compared with the existing literature, the innovation of this paper has not been well reflected”.
Answer. We appreciate the reviewer’s comments and suggestions. Comparison of the proposed method with existing methods has been carried out in the new version of the manuscript. In the Introduction, a new table has been presented in order to compare the proposed model with those based on machine learning and electrical circuits. The advantages of the proposed model over others models are emphasized. The following lines have been added in the Introduction: “In brief, according to the review of the state of the art, most of the models based on machine learning do not consider the dynamic characteristic of the polarization curve, likewise, its validation is not performed with a real FC data, and they have a high cost in their implementation. The proposed model overcomes these disadvantages, with high estimation accuracy”.
6. “Conclusion There is no quantitative data”.
Answer. Thanks for your comment. Quantitative data have been added in the conclusions. The data presented correspond to the results obtained using the proposed method, and those obtained from the comparison with other methods. In the new version of the manuscript we write the following sentences in the conclusions “The performance of the proposed model was carried out by computing the RMSE between the estimated voltage samples and the measured ones. The RMSE obtained for the first profile was 1 V, and 0.87 V for the second profile. These results are lower than those obtained with the Diffusive model which are 1.05 V for the first profile and 2.32 V for the second profile, and the Evolution strategy model which is 2.31 V for the second profile. Notice that, for the second profile, the error obtained with the proposed model is 62% lower than with the other models.”.
7. “Why are there two blank pages? Please check other errors carefully”.
Answer. Thanks for your comment. The document has been revised and blank pages have been removed.

Round 2
Reviewer 2 Report
Authors have addressed all my concerns.